# All-silicon quantum light source by embedding an atomic emissive center in a nanophotonic cavity

W. Redjem[1,6], Y. Zhiyenbayev[1,6], W. Qarony[1,6], V. Ivanov [2], C. Papapanos [1], W. Liu [2], K. Jhuria[2], Z. Y. Al Balushi [3,4], S. Dhuey[5], A. Schwartzberg [5], L. Z. Tan [5], T. Schenkel [2] & B. Kanté [1,4] ✉

Silicon is the most scalable optoelectronic material but has suffered from its inability to generate directly and efficiently classical or quantum light on-chip. Scaling and integration are the most fundamental challenges facing quantum science and technology. We report an all-silicon quantum light source based on a single atomic emissive center embedded in a silicon-based nanophotonic cavity. We observe a more than 30-fold enhancement of luminescence, a near-unity atom-cavity coupling efficiency, and an 8-fold acceleration of the emission from the all-silicon quantum emissive center. Our work opens immediate avenues for large-scale integrated cavity quantum electrodynamics and quantum light-matter interfaces with applications in quantum communication and networking, sensing, imaging, and computing.

Quantum science and technologies promise to revolutionize our societies[1,2]. In the search for the ideal quantum information processing platform, "scaling" is perhaps the most challenging question due to the fundamental but contradictory requirements for quantum systems to simultaneously be isolated and controllable from the environment in large arrays of interacting qubits[3,4]. Among many quantum information platforms ranging from superconducting qubits to trapped ions, photons play a fundamental role because they are necessary for future quantum networks to enable communication between distant quantum nodes[5–9]. Single photons have been generated from an extensive range of platforms, including quantum dots, color-centers in diamonds such as NV, SiV, and SnV, or defects in two-dimensional materials such as hBN[10–15]. The scaling challenge is currently being addressed using hybrid material platforms and metamaterials in which quantum light sources are optimized and integrated into more complex scalable systems, following the example of heterogeneous integration in the classical domain[16–20]. However, the challenge for integrating quantum devices is more significant than for classical systems because each interface allows losses and decoherence that need to be minimized. It is thus fundamental to minimize the number of interfaces by deeply integrating intrinsically scalable platforms.

Silicon is currently the most scalable optoelectronic material. Despite the lack of efficient classical light sources based on silicon, emissive centers have been observed in silicon since the end of the 1980s[21]. It is only during the last two years that single centers in silicon have been isolated[22–25]. Since then, emissive centers in silicon have been coupled to waveguides, and more recently, an ensemble of centers has been integrated into ring resonators[26–29]. Recently, efforts have been made to fabricate controllable such single emissive centers in silicon with high probability[30]. However, deterministic single-photon sources based on silicon emissive centers have remained elusive due to the lack of controlled manufacturing approaches and the complexity of materials interfaces after device fabrication. We report an all-silicon quantum light source based on an atomic emissive center in a silicon nanophotonic cavity. The manufacturing of the centers in silicon-on-insulator substrates, with controlled densities and preferential dipole

[1]Department of Electrical Engineering and Computer Sciences, University of California Berkeley, Berkeley, CA 94720, USA. [2]Accelerator Technology and Applied Physics Division, Lawrence Berkeley National Laboratory, Berkeley, CA 94720, USA. [3]Department of Materials Science and Engineering, University of California Berkeley, Berkeley, CA 94720, USA. [4]Materials Sciences Division, Lawrence Berkeley National Laboratory, Berkeley, CA 94720, USA. [5]Molecular Foundry, Lawrence Berkeley National Laboratory, Berkeley, CA 94720, USA. [6]These authors contributed equally: W. Redjem, Y. Zhiyenbayev, W. Qarony. ✉e-mail: bkante@berkeley.edu

orientations, increases their overlap probability with designed nanophotonic cavities. We demonstrate the successful alignment of a quantum defect and nanophotonic cavity dipole moments and tune the nanophotonic cavity to overlap its resonance with the zero-phonon line of the silicon-based quantum defect. We achieved a more than 30-fold enhancement of the luminescence intensity and an 8-fold acceleration of the single-photon emission rate. Our results open the door to large-scale integrated all-silicon quantum optics devices and systems for applications in quantum communication, sensing, imaging, and computing.

## Results

### All-silicon atom-cavity system

The proposed all-silicon atom-cavity system, presented in Fig. 1a, consists of a single defect in silicon embedded in a photonic crystal (PhC) defect cavity. The PhC cavity consists of three missing holes in a suspended triangular lattice of holes. The atomic defect, the G-center in silicon, is made of two substitutional carbon atoms (black spheres) bound to the same silicon self-interstitial (blue sphere). The manufacturing process starts with the implantation of carbon ($^{13}$C) with an energy of 36 keV in a commercial 230 nm thick silicon-on-insulator (SOI) wafer. The implantation is followed by electron beam lithography, dry etching, thermal annealing, and wet etching (see Supplementary Note 1 and 2). The rapid thermal annealing is an important step to thermally cure the broad luminescence from W-centers and G-centers induced by the dry etching process (see Supplementary Note 2). Secondary ion mass spectroscopy (SIMS) measurements indicate that the implanted carbon and the atomic centers created during the annealing process are located in the middle of the silicon layer (see Supplementary Note 2).

The dipole moment of the center is computed by density functional theory[31]. It is in the plane as indicated by the red arrow in the inset of Fig. 1a (see Supplementary Note 1). The G-center is one of a broad diversity of recently observed emissive centers in silicon and its electronic structure, presented in Fig. 1b, comprises a ground singlet state, a dark excited triplet state, and an excited singlet state[32]. The computed electromagnetic mode of the cavity for the transverse electric polarization is superimposed on the sketch of the PhC, evidencing the high confinement of the electromagnetic field in the region of missing holes in the triangular lattice[33]. This polarization matches the orientation of the atomic defect dipole moment. The deterministic positioning of atomic-scale defects in photonic cavities has been challenging for most platforms and has not yet been achieved for silicon-emissive centers. It requires not only the overlap of the quantum defect with highly confined optical modes but also the alignment of the dipole moments of the atom and the cavity. To increase the overlap probability in our platform, we first investigated the scalable manufacturing of single emissive centers with controllable densities and inhomogeneous broadening[34]. We identified an annealing time window below which only ensembles of centers are created and beyond which all single centers are destroyed (see Supplementary Note 1). We also find that shorter annealing time within that window minimizes the inhomogeneous broadening of the zero-phonon line (ZPL) of the quantum emitters, a critical requirement for overlapping the ZPL with a designed nanophotonic resonance to enhance light-matter interaction (see Supplementary Note 1). The controlled density and inhomogeneous broadening of quantum centers increase the probability of overlap with an array of finite-size photonic crystal cavities. We subsequently investigated the polarization response of created emissive centers, and a statistical analysis presented in the Supplementary Note 1 indicates a preferential orientation of the emitters in silicon. We then fabricated PhC cavities so that the dipole moments of the cavities and centers

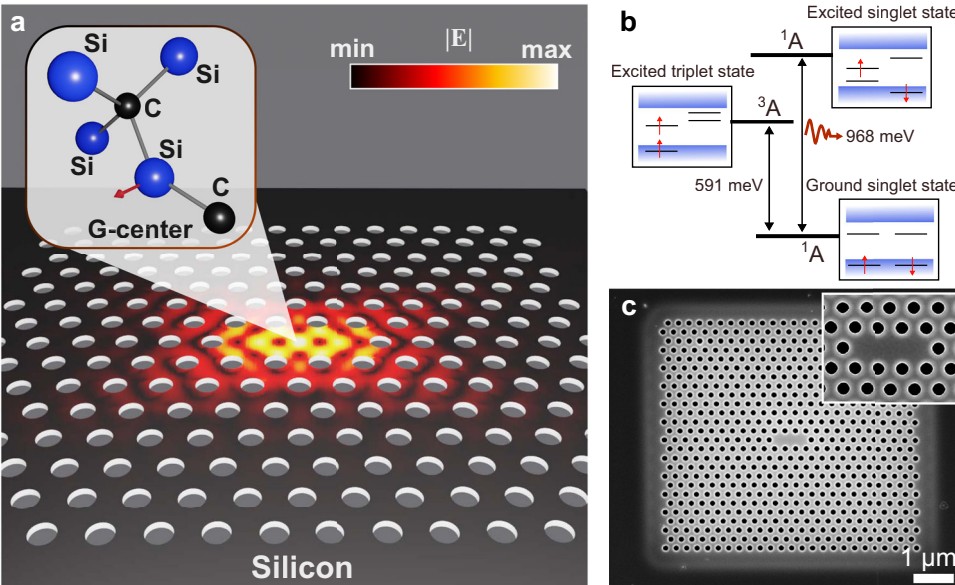

**Fig. 1 | A single atomic emissive center embedded in a silicon-photonic cavity.** **a** Silicon quantum interface with an "atomic defect" located within the "photonic defect" cavity of three missing holes in a triangular photonic crystal (PhC). The atomic defect is the G-center in silicon made of two substitutional carbon atoms (black spheres) bound to the same silicon self-interstitial (blue sphere). The red arrow indicates the direction of the dipole moment of the G-center. The G-center is one of a broad diversity of recently observed emissive centers in silicon, the most scalable opto-electronic material. The computed electromagnetic mode of the cavity is superimposed on the sketch of the PhC, evidencing the high confinement of the electromagnetic field in the region of missing holes in the triangular lattice. The cavity is fabricated so that its dipole moment and the dipole moment of the defect are colinear (see Supplement Note 1). The electric field strength peaks at the center of the cavity and exponentially decays in the bulk of the PhC. **b** Energy level diagram of the G-center in silicon comprising a ground singlet state, a dark excited triplet state, and an excited singlet state. The cavity can be tuned to be in resonance with the radiative transition between the excited and ground singlet states to enhance light-matter interaction. **c** Scanning electron microscope (SEM) image of a fabricated silicon-based atom-cavity system suspended in the air. The successful embedding of a single G-center in a photonic cavity involved a controlled sequence of fabrication steps using a commercial 230 nm thick silicon-on-insulator (SOI) wafer that is carbon implanted, followed by electron beam lithography, dry etching, thermal annealing, and wet etching (see Supplementary Note 2). The fabrication steps, compatible with standard complementary metal-oxide semiconductor (CMOS) processes, are optimized to increase the probability of single color-centers in cavities (see Supplementary Note 2).

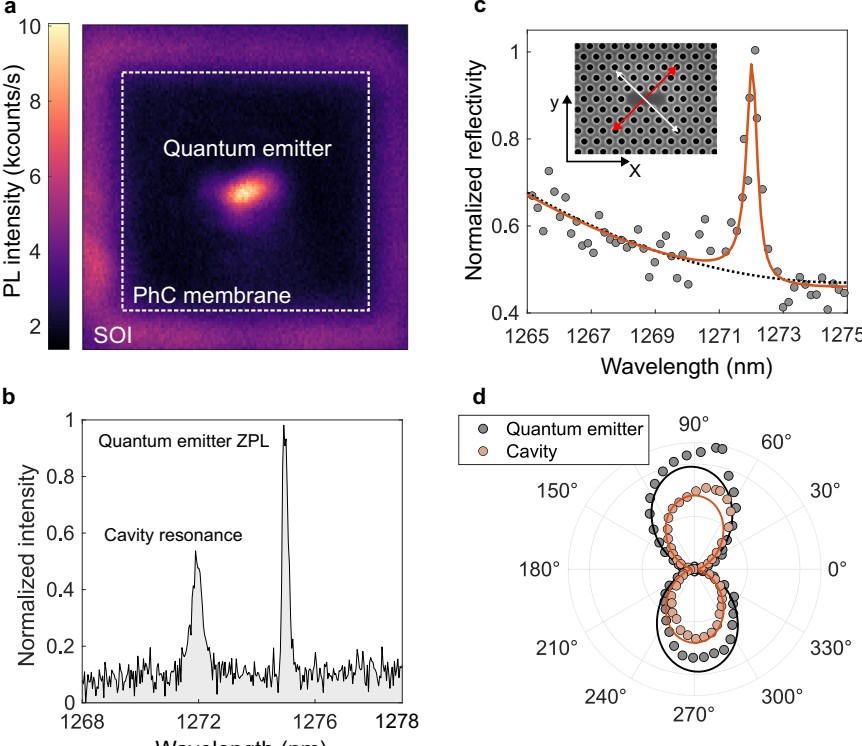

**Fig. 2 | Experimental characterization of the silicon-based quantum emitter and cavity. a** Photoluminescence (PL) raster scan of a device with a single emitter in the cavity at a temperature of 4K. The boundary of the finite photonic crystal (PhC) is indicated by the dashed white line and the suspended PhC is surrounded by the silicon-on-insulator (SOI) wafer. The photoluminescence signal shows bright emission from a color-center within the boundaries of the cavity. **b** The photoluminescence of the photonic device exhibits a sharp peak at ~1275 nm and a blue-shifted broader peak at ~1272 nm, corresponding to the zero-phonon line of the color-center and the cavity resonance, respectively. **c** Reflectivity of the photonic crystal cavity obtained by resonant scattering measurements. The cavity is illuminated with a linearly polarized white light source (white arrow) at 45° with respect to the cavity axis that is along the X-direction. The polarized signal perpendicular to the excitation is collected (red arrow) to probe the cavity mode. **d** The polarization diagram of the cavity mode detuned from the ZPL is shown in orange. The polarization diagram of a quantum emitter alone is shown in black. The polarizations agree well with a dipolar model (solid lines) and are well aligned (see Supplementary Note 1).

align. Figure 1c presents a scanning electron microscope (SEM) image of a fabricated silicon-based atom-cavity system. The inset presents the cavity with a mode volume of $0.66(\lambda_{cav}/n)^3$. The successful embedding of a single center in a cavity involved a controlled sequence of CMOS-compatible fabrication steps (see Supplementary Note 2).

## Characterization of the atom-cavity system

Figure 2a presents the photoluminescence (PL) raster scan of a device at a temperature of 4 K with bright emission from a color-center within the boundaries of the cavity. The dashed white line indicates the boundary of the finite PhC, and the suspended PhC is surrounded by the silicon-on-insulator (SOI) wafer. The photoluminescence of the photonic device, presented in Fig. 2b, exhibits a sharp peak at ~1275 nm and a blue-shifted broader peak at ~1272 nm, corresponding to the ZPL of the color-center and the cavity resonance, respectively. The cavity's photoluminescence originates from the broad spectrum of the background centers. The cavity is further characterized in reflectivity using resonant scattering measurements in Fig. 2c. The cavity is illuminated with a linearly polarized white light source (white arrow) at 45° with respect to the cavity axis that is along the X-direction. The signal polarized perpendicular to the excitation is collected (red arrow) to probe the cavity mode, and a resonance is observed at ~1272 nm, in perfect agreement with the PL measurement. The reflectivity is fitted with a Fano resonance line shape giving an intrinsic quality factor (Q) of 3209. The experimental value is comparable to the theoretical Q of 6000 and the discrepancy is attributed to fabrication imperfections. Figure 2d presents the polarization diagram of the cavity mode

detuned from the ZPL in orange and the polarization diagram of a quantum emitter alone in black. The polarizations agree well with a dipolar model (solid lines) and have been successfully aligned.

In Fig. 3a, the spectrum of the quantum emitter over a broad range of energy shows the zero-phonon line (ZPL) of the silicon emissive center and its phonon side band. Figure 3b presents the spectrum of the quantum emitter using a high-resolution grating. The ZPL is located at 972.43 meV and has a linewidth of 6.8 GHz (obtained after deconvolution with the spectrometer response function). To demonstrate that the bright emission from the middle of the cavity corresponds to a single emissive center, we performed quantum coherence measurements of the emitter in the cavity. Autocorrelation measurements, shown in Fig. 3c, are performed using a Hanbury-Brown and Twiss interferometer with superconducting nanowire single-photon detectors (see Supplementary Note 3). The second-order correlation measurements of the emission from the cavity under continuous excitation exhibit an antibunching, confirming the successful spatial overlap of a single silicon emissive center with the nanophotonic cavity with an antibunching at zero delay $g^2(0) = 0.30 \pm 0.07$. The value at zero delay is mainly limited by the emission from background centers. Autocorrelation measurements under pulsed excitation at a repetition rate of 10 MHz are presented in Fig. 3d.

## Cavity enhanced single-emissive center in silicon

The enhancement of the single center in the cavity requires spatial and spectral overlap. Spatial overlap was achieved in Fig. 2 and Fig. 3. To achieve spectral overlap, the nanophotonic cavity is tuned using cycles

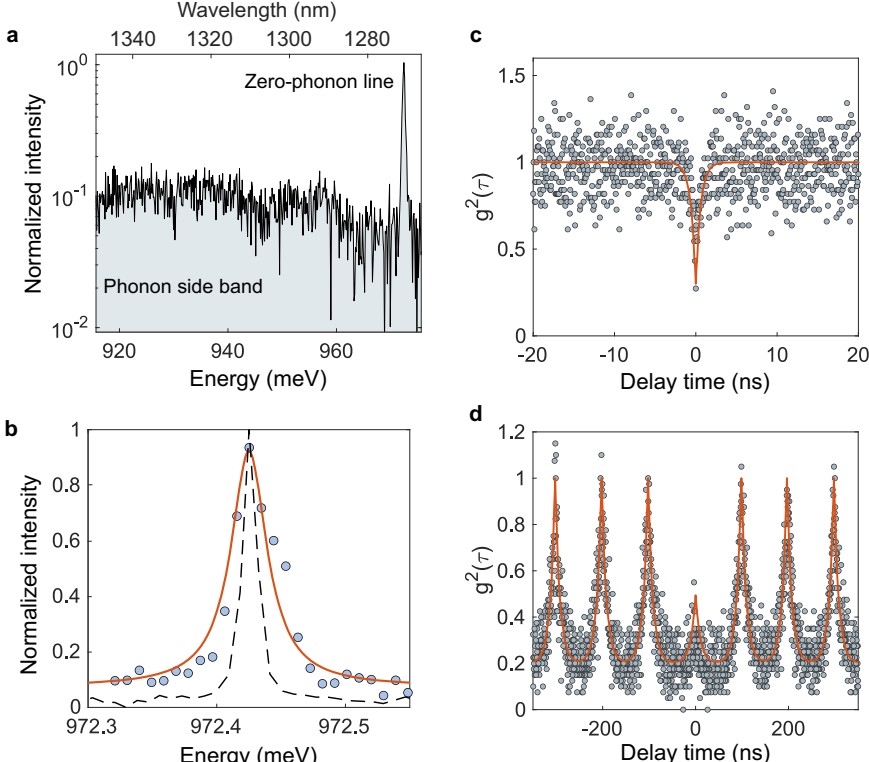

**Fig. 3 | Quantum coherence measurements of the emitter in the cavity.**
**a** Spectrum of the quantum emitter over a broad range of energy showing the zero-phonon line (ZPL) of the silicon emissive center and its phonon side band.
**b** Spectrum of the quantum emitter using a high-resolution grating. The ZPL is located at 972.43 meV and has an intrinsic linewidth of 6.8 GHz. The dashed line corresponds to the instrument limit. **c** Second order autocorrelation measurements of the emission from the cavity under continuous excitation. The

antibunching at zero delay confirms the successful spatial overlap of a single silicon emissive center with the nanophotonic cavity with an antibunching at zero delay $g^2(0) = 0.30 \pm 0.07$. **d** Second order autocorrelation measurements under pulsed excitation at a repetition rate of 10 MHz. Autocorrelation measurements are performed using a Hanbury-Brown and Twiss interferometer with superconducting nanowire single-photon detectors (see Supplementary Note 3).

of argon gas injection. The injected gas condensates at the surface of the PhC and modifies the effective index of cavity mode tuning the resonance wavelength of the cavity that is shifted from ~1269 nm to ~1275 nm. In Fig. 4a, as the cavity resonance is shifted towards the ZPL of the quantum center, the photoluminescence is enhanced to reach a maximum at ~1275 nm, where the spectral overlap is achieved. The ZPL intensity as a function of the cavity detuning shows an enhancement larger than 30 achieved on resonance (Fig. 4b). For cavity detuning varying from $\delta = 2.40$ nm to $\delta = 0.00$ nm, the excited lifetime shortens from 53.6 ns to 6.7 ns. An 8-fold reduction in the lifetime is experimentally observed when the overlap is achieved compared to the off-resonance case. Light-matter interaction in cavities is usually quantified using the Purcell factor ($F_p$) that measures the decay rate enhancement of the atom from free space to the cavity ($\gamma_{cav} = F_p\gamma_0$). It can be estimated by $F_p = (\tau_{bulk}/\tau_{on} - \tau_{bulk}/\tau_{off})/\eta$ where $\tau_{bulk}$ is the lifetime of a quantum emitter outside the photonic crystal (dark yellow dots in Fig. 4c), and $\tau_{off}$ is the lifetime for a detuning of 2.4 nm. The lifetime measured at off-resonance is slightly longer than the one in the bulk because of the reduced density of state in the PhC gap[35].

## Discussion

We further confirmed that the Purcell enhancement is due to the cavity-emitter interaction and present the detuning-dependent Purcell factors in the supplementary Note 4. The percentage of photons emitted at the ZPL wavelength of the emitter (Debye-Waller) is $\eta = 15\%$, which was measured by comparing the count rate with and without the ZPL bandpass filter. The experimental Purcell factor of the defect cavity is $F_p \sim 29.0$. The coupling efficiency of the center to the cavity mode ($\beta$ factor) can be estimated by $1/\tau_{on}/[1/\tau_{on} + 1/\tau_{off}]$, yielding a

value of $\beta \sim 89\%$. The emission from the ZPL of not enhanced centers is about 700 c/s. We measured a Purcell enhancement of over 30 with a count rate of 20000 c/s. Enhancement of single centers was observed in several other cavities (see Supplementary Note 2). A systematic enhancement of single emitters in cavities was observed but there was not a direct correlation between the enhancement and the Q-factor because the positioning of centers in cavities is probabilistic. The lifetime reduction and Purcell acceleration observed in our work for a single center indicates a close to unity quantum efficiency. The mechanism leading to the formation of various single G-centers from the ensemble is currently an open question[36].

We thus reported an all-silicon quantum light source based on an atomic emissive center in a silicon-based nanophotonic cavity. The quantum center is manufactured directly in silicon using a sequence of complementary metal-oxide semiconductor (CMOS) compatible nanofabrication steps that simultaneously control the implantation depth, the density, the inhomogeneous broadening, and the dipolar orientation of the emitters. The control of these parameters enables the successful embedding of the atomic defect in a silicon photonic crystal defect nanocavity. The nanocavity was spectrally tuned using gas condensation to overlap its resonance with the zero-phonon line of the quantum emitter, enhancing light-matter interaction. Our work will enable all-silicon quantum optics interfaces for scalable and integrated quantum optoelectronics.

## Methods
### Design
The photonic crystal (PhC) consists of a periodic arrangement of holes in a triangular lattice. The PhC has a complete gap centered around

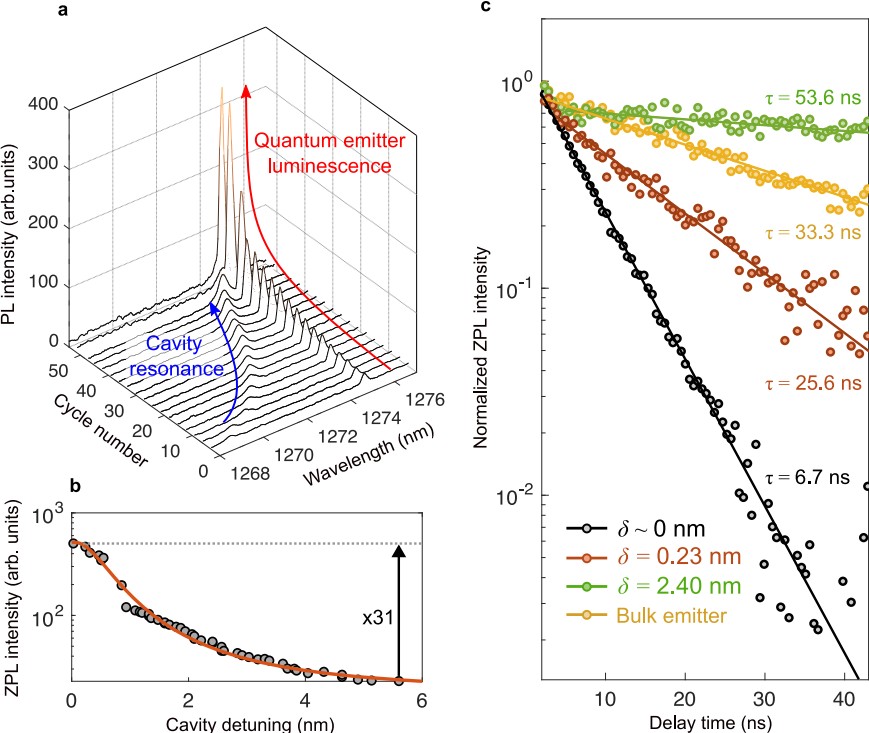

**Fig. 4 | Spectral tuning of the nanocavity and enhanced atom-cavity interaction. a** The enhancement of the single center in the cavity requires spatial and spectral overlap. Spatial overlap was achieved in Fig. 3 and spectral overlap is achieved here by tuning the nanophotonic cavity. Cavity tuning as a function of the argon gas injection cycle. Gas injection modifies the effective index of cavity mode and tunes the resonance wavelength of the cavity that is shifted from ~1269 nm to ~1275 nm. As the cavity resonance is shifted towards the ZPL of the quantum center, the photoluminescence is enhanced to reach a maximum at ~1275 nm, where the spectral overlap is achieved. **b** Zero-phonon line intensity as a function of the detuning of the cavity. An enhancement larger than 30 is achieved on resonance. **c** Excited lifetime for cavity detuning of δ = 2.40 nm, δ = 0.23 nm, and δ = 0.00 nm.

The lifetime of emitted photons shortens from 53.6 ns to 6.7 ns when the detuning between the cavity and emitter is decreased. The instrument response function of the pulsed laser has also been provided in the Supplementary Note 3. An 8-fold reduction in the lifetime is experimentally observed when the overlap is achieved compared to the off-resonance case. These results constitute the first all-silicon quantum light source using a silicon emissive center in a cavity, and the center can be further accelerated by designing cavities with higher quality factors as well as more deterministic positioning methods to further improve the emitter-cavity spatial overlap. The results will enable all-silicon quantum optics interfaces with silicon-emissive centers for scalable quantum optics.

1300 nm with a width of 284.3 nm. The period of the lattice is 346.5 nm, the radius of the air hole is $R = 96$ nm, and the silicon membrane thickness is $h = 230$ nm. To form the nanophotonic cavity, three holes are removed along the $\Gamma$-$K$ direction of the PhC. The missing holes create localized states at frequencies within the photonic band gap. As a result, the modes occupying these localized states exhibit high-quality factors and small mode volumes.

## Fabrication
A silicon-on-insulator (SOI) wafer with a 230 nm silicon layer is carbon implanted at 36 keV with a fluence of $4 \times 10^{13}$ cm$^{-2}$. The chip is exposed by electron-beam lithography and etched by an inductively coupled plasma dry etching process, and annealed using a rapid thermal annealing processor at 1000 °C for 5 s. Finally, the membranes are suspended using wet etching process.

## Characterization
The experiment is conducted at a temperature of 4 K, which is achieved using a closed-cycle cryogenic chamber. To excite the photoluminescence, we used either green lasers at 532 nm which can be either continuous or pulsed. The PL signal is separated from the pump beam using a dichroic mirror with a cut-off wavelength of 1000 nm. A long-pass filter at 1250 nm is used to increase the signal-to-noise ratio. A combination of a steering mirror and lenses is used for raster scanning the sample. Superconducting nanowire single-photon detectors (SNPDs) are used for photon counting and time-correlation measurements. The signal spectrum is analyzed using an InGaAs camera out-

couple to a grating spectrometer. For reflectivity measurements, we used a supercontinuum white light source along with a polarizer and beam splitter. A half-wave plate is used to rotate the polarization of the incident light to match the polarization of the nanophotonic cavity.

## Reporting summary
Further information on research design is available in the Nature Portfolio Reporting Summary linked to this article.

## Data availability
The data that support the plots within this paper and other findings of this study are available from the corresponding author upon request.

## Code availability
The computer codes that support the plots within this paper and other findings of this study are available from the corresponding author upon request.

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

## Acknowledgements

This research was supported by the Moore Inventor Fellows programme, the NSF QLCI programme through grant number OMA-2016245, and the NSF QuIC-TAQS award 2137645. It was partially supported by the Office of Naval Research (ONR) grant N00014-20-1-2723, the ONR grant N00014-22-1-2651, the U.S. Army Research Office (ARO) grant W911NF2310027, and the Bakar Fellowship at UC Berkeley. The work was partially performed at the UC Berkeley Marvel Nanofabrication Laboratory and the Molecular Foundry at the Lawrence Berkeley National Laboratory. T.S. and L.Z.T. acknowledge support from the Office of Science, Office of Fusion Energy Sciences, of the U.S. Department of Energy, under Contract No. DE-AC02-05CH11231. L.Z.T. was supported by the Molecular Foundry, a DOE Office of Science User Facility supported by the Office of Science of the U.S. Department of Energy under Contract No. DE-AC02-05CH11231. This research used resources of the National Energy Research Scientific Computing Center; a DOE Office of Science User Facility supported by the Office of Science of the U.S. Department of Energy under Contract No. DE-AC02-05CH11231.

## Author contributions

W.R., T.S., and B.K. conceived the project. Y.Z. designed the structures and performed the simulations. W.Q. fabricated the devices with the support of S.D. and A.S.; W.R and Y.Z. performed the measurements. C.P. and Z.Y.A.B. contributed to gas tuning. W.L. and K.J. contributed to the implantation of the samples. V.I. and L.Z.T. performed the DFT calculations. B.K. supervised the project. All authors contributed to the discussions and W.R., Y.Z., W.Q., and B.K. wrote the manuscript.

## Competing interests

The authors declare no competing interests.
