## [Peer Review File · Nature Communications]

All-silicon quantum light source by embedding an atomic emissive center in a nanophotonic cavityREVIEWER COMMENTS

Reviewer #1 (Remarks to the Author):

Color centers in the solid state are promising platforms for quantum applications including quantum sensing and quantum communication. Moving towards a scalable technology, it is desirable to integrate these color centers into nano-devices. A majority of works on this front are focused on color centers in diamond, but device integration of diamond color centers is hard due to the challenges in diamond fabrication. Recently, a significant amount of interest has shifted to color centers in silicon, with the demonstration of telecom-band single photon emitters in silicon [Nature Electronics 3, 738–743 (2020)], and excellent spin properties of T centers in silicon [PRX Quantum 1, 020301 (2020)]. Due to the technological importance of silicon for photonics, and semiconductor industry, developing understanding and photonics platform for color centers in silicon would be of great importance. Some prior works in this direction includes waveguide integrated G-/T- centers [arxiv.2202.02342, arxiv.2209.14260, 2211.09305], and Purcell enhanced emission from ensemble G centers in a ring resonator [arxiv.2210.05485].

The manuscript by Redjem et al demonstrates Purcell enhancement of a single G center in silicon inside a 2D photonic crystal cavity. They observe a 30-fold enhancement in PL and an 8-fold decrease in the emission lifetime. This is enabled by aligning the dipole orientation to the cavity dipole and understanding the formation condition of single G centers. The experimental results would be of interest to the silicon color center community. I think the manuscript is in principle suitable for the audience of Nature Communications. However, the authors remain somewhat short on some scientific questions, and I would like to hear the response from the authors. Here are some major comments:

1. The authors described and demonstrated a rather complicated method to generate single G centers by first generating an ensemble of G centers and then controlling the annealing time to reduce the density down to the single-center level. Is it not possible to simply do a lower density implantation?
2. Following #1, I am a bit confused about the formation of single G centers via rapid thermal annealing. It seems that the authors observed a much broader inhomogeneous distribution for the annealed single centers compared to the unannealed ensemble centers. Is there any explanation/hypothesis on this observation? Normally a broader distribution would come from high strain in the substrate – but I would expect annealing to repair crystal damage (therefore reduce strain). Did the author observe any obvious difference between the ensemble centers and the single centers (brightness, lifetime, etc)? For example, based on Figure S1, it seems that the single centers formed after annealing is much brighter compared to the ensemble centers (presumably also a much higher density).
3. The main results are based on one cavity-coupled G center. Device yield is a very important factor for the technique to be scalable. Are similar measurements performed on other devices? The authors showed the distribution of G-center ZPL and cavity resonance in Fig. S10. It would be nice if the authors could provide similar statistics on cavity Q and ZPL enhancement/lifetime reduction of other cavity coupled G centers.
4. I noticed that there is some inconsistent observation on the properties of G centers in literatures. For example, in [arxiv.2202.02342, arxiv.2211.09305, arxiv.2210.05485] (which the authors also cited), the lifetime of G centers was reported to be 4 to 8 ns, and in [arxiv.2210.05485] the quantum efficiency was estimated to be below 10%. However, in this work, the authors observe single-center lifetimes (no cavity reduction) >30 ns, and a significant reduction of the lifetime with cavity enhancement. This means the quantum efficiency of the single G center in this work is really high. The authors should provide some hypothesis on this discrepancy as this would be very important for future quantum applications of G centers.

Some minor comments that I hope the authors could also address:

1. How was the location of the G-center aligned to the cavity region? The authors mentioned deterministic positioning is important: "The deterministic positioning of atomic-scale defects in photonic cavities has been challenging for most platforms and has not yet been achieved for silicon-emissive centers. It requires not only the overlap of the quantum defect with highly confined optical modes but also the alignment of the dipole moments of the atom and the cavity". However, based on the fabrication procedure it seems the spatial alignment is probabilistic. Please clarify.
2. In PL measurement, the cavity resonance is observed as a broad resonance at 1272 nm. Is this from some broadband fluorescence from silicon? Please clarify.
3. For the measured ZPL linewidth of 8.3 GHz, is this from the instrument limit or the intrinsic single center linewidth?
4. Following pulsed excitation measurement of G center, the authors claimed on demand single photon generation "Autocorrelation measurements under pulsed excitation at a repetition rate of 10 MHz are presented in Fig. 3d and they demonstrate on-demand single-photon generation from the all-silicon platform". However, two important criteria for on-demand single photons are brightness and purity, which the authors did not discuss. Therefore, I think the authors should remove this claim. In addition, throughout the text, the authors did not mention the exact count rate of the Purcell-enhanced G center.
5. For g_2 measurements, the baseline at non-zero delay is fitted to exactly 1. How was the g_2 data normalized? I would expect some bunching from the shelving state (3A state shown in Fig 1B). In addition, the timescale in Fig 3c seems very fast compared to the 50 ns lifetime. Was the measurement performed with a very high excitation power? Please clarify.
6. For SIMS measurement (Fig. S7), there are two maximum of carbon concentration at around 0 nm and around 210 nm. Does the author know the origin of these local maximum? Could these be from carbon diffusion during annealing?

Reviewer #2 (Remarks to the Author):

The authors integrate G-center in silicon photonic crystal cavities and obtain 30-fold Purcell enhancement and 8-fold lifetime reduction of the ZPL single-photon emission at 1275 nm. The results are exciting because of the scalability of the silicon on insulator approach and the authors calibrated the annealing process to optimize single emitter generation and minimize spectral broadening. It is an important work, however, I do not find that it has innovative enough aspects for publication in Nature Communications, but would rather expect to read these findings in Nano Letters or ACS Photonics. Earlier this year Purcell enhancement of G-center ZPL in microrings has been reported in Applied Physics Letters, for example. There is also exciting work on indistinguishability of photons from T-centers and spin control of defects in silicon.

A few comments on the manuscript:

- the temperature of the experiment is missing
- 'quantum photon' is an unusual term since photon is already a quantum of light, perhaps use 'single-photon' or 'quantum light' instead

Reviewer #3 (Remarks to the Author):

The paper of Redjem et al. reports integration of single G-centers with a 2D photonic crystal cavity to utilize cavity-enhanced light-matter interactions, achieving 30-fold luminescence enhancement and 8-fold emission acceleration. Although using nanophotonic cavity to Purcell-enhance the fluorescence had been demonstrated in multiple solid-state atomic systems and single G-centers in waveguide had been observed before, there is still incremental novelty to analyze single G-centers coupled with the cavity. In large parts, the paper is well-written with well-presented data and convincing conclusions. Thus, I can recommend publication in Nature Communications if the authors can address following concerns and questions:

1. The ZPL of the single G-center seems to have a rather wide distribution (e.g., Fig. S4, s10) after the generation process including the ion implantation and annealing. For developing quantum sources, e.g., indistinguishable photons from two G-centers, the wide inhomogeneous distribution can be problematic. Can authors comment on the origin of this wide ZPL distribution, and potential ways to mitigate/decrease it?
2. Another important measure of the quantum light source is the photon extraction efficiency, which is a common issue for many solid-state emitters. With the coupling regime authors used, what is the photon extraction efficiency?
3. It's quite misleading when authors mentioned (in line 56, 57) they can manufacture G-centers with controlled dipole orientations. From Figure S6, G-center dipole orientation after fabrication is random, but with preferences, which enables cavity polarization to match with the G-center dipole. Authors may want to rephrase those sentences to clear this confusion in the main text.
4. What is the average distance between G-centers in the SOI sample? How many G-centers will locate in the mode volume of the L3 cavity? Does the large $g^2(0)$ being limited by the detection noise (e.g., dark counts) or by background G-center emissions?
5. In the G-center decay measurements, what is the instrumentation response function? Authors may just want to add that in Fig. 4c.
6. To verify the fluorescence decay enhancement is indeed coming from the emitter-cavity interactions, authors may want to present detuning-dependent Purcell factor data and fit it with the Lorentzian (Equation 2 in SI), and to check whether the fitted linewidth matches with the cavity.
7. The deposition of the Argon ice seems to decrease the cavity Q factor (i.e., increase the cavity loss rate κ) as shown in Fig. S12c. Can authors comment on this, e.g., whether this comes from increased scattering or perturbation of the band structure? This increasing κ may cause complications on the fitting mentioned in the previous question.

Referee #1 (Remarks to the Author):

Color centers in the solid state are promising platforms for quantum applications including quantum sensing and quantum communication. Moving towards a scalable technology, it is desirable to integrate these color centers into nano-devices. A majority of works on this front are focused on color centers in diamond, but device integration of diamond color centers is hard due to the challenges in diamond fabrication. Recently, a significant amount of interest has shifted to color centers in silicon, with the demonstration of telecom-band single photon emitters in silicon [Nature Electronics 3, 738–743 (2020)], and excellent spin properties of T centers in silicon [PRX Quantum 1, 020301 (2020)]. Due to the technological importance of silicon for photonics, and semiconductor industry, developing understanding and photonics platform for color centers in silicon would be of great importance. Some prior works in this direction includes waveguide integrated G-/T- centers [arxiv.2202.02342, arxiv.2209.14260, 2211.09305], and Purcell enhanced emission from ensemble G centers in a ring resonator [arxiv.2210.05485].

The manuscript by Redjem et al demonstrates Purcell enhancement of a single G center in silicon inside a 2D photonic crystal cavity. They observe a 30-fold enhancement in PL and an 8-fold decrease in the emission lifetime. This is enabled by aligning the dipole orientation to the cavity dipole and understanding the formation condition of single G centers. The experimental results would be of interest to the silicon color center community. I think the manuscript is in principle suitable for the audience of Nature Communications. However, the authors remain somewhat short on some scientific questions, and I would like to hear the response from the authors. Here are some major comments:

Response of the authors to Referee 1

We would like to thank the reviewer for finding that the “manuscript is in principle suitable for the audience of Nature Communications.” Additionally, we thank the reviewer for the quality of his/her comments towards improving the manuscript and acknowledging that the results “would be of interest to the silicon color center community.” We address the comments of the reviewer in what follows.

Referee #1 (Remarks to the Author):

1. The authors described and demonstrated a rather complicated method to generate single G centers by first generating an ensemble of G centers and then controlling the annealing time to

reduce the density down to the single-center level. Is it not possible to simply do a lower density implantation?

Response of the authors to Referee 1

We thank the reviewer for this question. The referee is correct and lower density of carbon implantation can be used to form single centers directly. It has been demonstrated in the literature [Hollenbach et al. *Opt. Express* 28, 26111-26121 (2020) ref 24 in the main text] that single centers can be formed with a fluence as low as 10^{10} C/cm² and low energy of 5 keV. For such conditions, single centers are formed without the need for annealing. However, the centers are created at 20 nm from the top surface of the silicon. In our work, we used the energy of 36 keV to create the centers at a depth of ~100 nm (almost in the middle of the thin silicon slab of 230 nm), where the optical field in the cavity is maximum. For 36 keV energy, we were able to create only an ensemble of G-centers even for low fluences such as $<10^{10}$ cm².

Usually, an ensemble of luminescent centers, such as W and G, are created during the dry etching of the silicon material [Weber, J. et al. *Appl. Phys. A* 41, 175–178 (1986)]. The challenge is that the ensemble gives a very large photoluminescence background that makes the detection of single centers challenging. Even after chemical etching of only about 200 Angstrom of the surface, no G-line could be detected [Weber, J. et al. *Appl. Phys. A* 41, 175–178 (1986)]. Even though the technique of metal-assisted chemical etching would prevent the etching-assisted creation of centers [Journal of Applied Physics 132, 033101 (2022)], we opted for more conventional fluorine-based processes widely used in silicon photonics. The annealing step is thus critical to thermally destroy the centers created at the surface by the dry etching process and control the density and inhomogeneous broadening [Opt. Express 31, 8352-8362 (2023)].

Following this question from the reviewer, we have added a sentence to the manuscript. Please see the highlighted sentence in page 3, first paragraph **“Rapid thermal annealing is an important step to thermally cure the broad luminescence from W-centers and G-centers induced by the dry etching process (see Supplementary Information).”**

We also added a sentence in the supplementary information Section 2.2 the following:

Usually, an ensemble of luminescent centers, such as W and G, are created during the dry etching of the silicon material [4]. We perform rapid thermal annealing to thermally destroy the ensembles and to form single-color centers with a high signal-to-background noise ratio in carbon-implanted samples.

Referee #1 (Remarks to the Author):

2. Following #1, I am a bit confused about the formation of single G centers via rapid thermal annealing. It seems that the authors observed a much broader inhomogeneous distribution for the annealed single centers compared to the unannealed ensemble centers. Is there any explanation/hypothesis on this observation? Normally a broader distribution would come from high strain in the substrate – but I would expect annealing to repair crystal damage (therefore reduce strain). Did the author observe any obvious difference between the ensemble centers and the single centers (brightness, lifetime, etc)? For example, based on Figure S1, it seems that the single centers formed after annealing is much brighter compared to the ensemble centers (presumably also a much higher density).

Response of the authors to Referee 1

We thank the reviewer for this question. After implantation and prior to annealing, the sample shows a high fluence of luminescence which we refer to as “ensemble.” The density of Carbon-Carbon (CC) pairs is high enough that we always observe many CC’s within the field of view. As the sample is annealed, many color centers “dissolve,” and the density decreases to the point where individual centers can be isolated. However, we would like to emphasize that in our study, the ensemble of G-centers is not just a collection of single G-centers because of the following observations:

- **The single center's distribution peak is blue-shifted from the G-line of the ensemble. When varying the annealing time, we didn't observe a continuous shift of the ZPL peak distribution.**
- **The inhomogeneous broadening of the single centers is much larger than the one for the ensemble.**
- **The lifetime of the ensemble of G-centers is 6 ns, while single centers have a lifetime of 35 ns.**
- **For the same excitation power and same beam size, the single center is almost twice as bright as the ensemble (see figure R1 below).**

To understand the origin of the broadening and shifting of the ZPL statistics, we performed ab initio calculations and computed the photoluminescence spectrum of the G-centers under strain [please see Opt. Express 31, 8352-8362 (2023)]. We found that the volume expands by 1.22% for the isolated G-center compared to the volume in the presence of a nearby G-center. According to our calculations, the expansion of the lattice corresponding to tensile strain would lead to a blue shift of the ZPL. Thus, the lattice is under high compression after implantation, and annealing releases some of the strain. However, since we used a very short annealing time, the lattice is not fully healed, and annealing would create more local disorder.

This is due to the fact that the annealing of silicon results in several microscopic processes which can affect the defect emission locally. For instance, we can have the release of Carbon interstitials from a G-center, the release of silicon interstitials and the formation of vacancy near a G-center, and/or incorporation of carbon interstitials into substitutional lattice positions near a G-center.

In the three processes discussed above, the silicon lattice locally contracts by a percent or less, leading to an overall “blue shift” of the emission. Moreover, the magnitude of the induced local strain is highly dependent on the distance between the G-center and the location of the microscopic process causing the strain (see Supplement). We notice that the strain can vary significantly depending on the location of the vacancy and may cause the observed large inhomogeneous broadening in the emission of single centers [Opt. Express 31, 8352-8362 (2023)].

Figure R1: Photoluminescence spectrum of an ensemble (black) and a single center (red) under the same excitation power and temperature.

Following this question from the reviewer, we have added Figure R1 to the supplement, and we also have added the above paragraph to the supplement (section 1.1) with Figure R1.

Following the reviewer question, we added the sentence below to the manuscript in page 3, second paragraph: **“To increase the overlap probability in our platform, we first investigated the scalable manufacturing of single emissive centers with controllable densities and inhomogeneous broadening [Opt. Express 31, 8352-8362 (2023)].”**

Referee #1 (Remarks to the Author):

3. The main results are based on one cavity-coupled G center. Device yield is a very important factor for the technique to be scalable. Are similar measurements performed on other devices? The authors showed the distribution of G-center ZPL and cavity resonance in Fig. S10. It would be nice if the authors could provide similar statistics on cavity Q and ZPL enhancement/lifetime reduction of other cavity coupled G centers.

Response of the authors to Referee 1

We agree with the referee that achieving high device yield is crucial for a scalable technique. Following the reviewer’s question, we performed additional experiments using different devices. Figure R2a represents photoluminescence spectra of cavities containing single centers on-resonance, along with their corresponding Q-factor values in Figure R2b. Notably, the black spectrum at 1271 nm represents a single center located in the silicon bulk, and we observed a 2- to 13-fold enhancement in the zero-phonon line (ZPL) intensity with respect to the single defect located outside of the cavity (bulk). Figure R2b shows the distribution of Q-factors, exhibiting an average value of 2525. Furthermore, Figure R3 displays the raster scan images and the lifetime reduction of two other cavities (devices 2 and 3) in addition to one presented in the main text (device 1). As shown in the table below (Table R1), we measured different Purcell factors. However, we did not observe a direct correlation between the Q-factor and the Purcell factor because the positioning of centers in cavities is probabilistic.

Figure R2. Statistics of the quantum emitters enhanced via the cavity interaction. a, Distribution of ZPLs enhanced by the cavity resonance. b, Distribution of the quality factors as a function of PL enhancement compared to the defect in the bulk.

Figure R3. Devices with enhanced and accelerated single photon emission. a,c, Raster scan images of the cavities (device 2 and 3) with a silicon defect located in the middle of the cavity region. b,d, Lifetime reduction of the emission in device 2 and 3 respectively.

Table R1: Table summarizing the Purcell factor/enhancement of three cavities containing a single center.

	Lifetime ON (ns)	Lifetime OFF (ns)	Purcell factor	β factor, %
Device 1 (main text)	6.7	53.6	29.0	89
Device 2	6.2	23.5	26.1	79
Device 3	21.1	46.9	5.7	69

Following the referee's remark, we have added Figures R2 and R3 in the supplementary section 2.5. We also modified the main manuscript in page 5, just before the conclusion and added:

Enhancement of single centers was observed in several other cavities (see Supplementary Information). We did not observe a direct correlation between the Q-factor and the Purcell factor because the positioning of centers in cavities is probabilistic.

Referee #1 (Remarks to the Author):

4. I noticed that there is some inconsistent observation on the properties of G centers in literatures. For example, in [arxiv.2202.02342, arxiv.2211.09305, arxiv.2210.05485] (which the authors also cited), the lifetime of G centers was reported to be 4 to 8 ns, and in [arxiv.2210.05485] the quantum efficiency was estimated to be below 10%. However, in this work, the authors observe single-center lifetimes (no cavity reduction) >30 ns, and a significant reduction of the lifetime with cavity enhancement. This means the quantum efficiency of the single G center in this work is really high. The authors should provide some hypothesis on this discrepancy as this would be very important for future quantum applications of G centers.

Response of the authors to Referee 1

We thank the reviewer for this comment. There is an ongoing debate about the discrepancies observed from different groups in terms of the linewidth, emission wavelength, and excited lifetime of single G-centers. A hypothesis was recently proposed by [Baron et al. Appl. Phys. Lett. 121, 084003 (2022)] regarding two types of single G-centers. The first one has the same structure and photoluminescence spectrum as the ensemble and a short lifetime (<6ns). The second type has a perturbed conformation with respect to the G-center structure which would correspond to the defect we reported in our work. They have a larger inhomogeneous broadening and a longer lifetime but a high quantum efficiency. Identifying this atomic configuration would require further experimental and theoretical investigations.

Following this question from the reviewer, we have added a sentence in the manuscript (see the paragraph just before the conclusion):

The lifetime reduction and Purcell acceleration observed in our work for a single center indicates a close to unity quantum efficiency. The mechanism leading to the formation of various single G-centers from the ensemble is currently an open question [35].

Referee #1 (Remarks to the Author):

Some minor comments that I hope the authors could also address:

1. How was the location of the G-center aligned to the cavity region? The authors mentioned deterministic positioning is important: “The deterministic positioning of atomic-scale defects in

photonic cavities has been challenging for most platforms and has not yet been achieved for silicon-emissive centers. It requires not only the overlap of the quantum defect with highly confined optical modes but also the alignment of the dipole moments of the atom and the cavity". However, based on the fabrication procedure it seems the spatial alignment is probabilistic. Please clarify.

Response of the authors to Referee 1

We thank the reviewer for this question. To achieve high coupling, the center needs to be located near the maxima of the optical field, and the associated dipole needs to be collinear to the polarization of the resonator. The overlap in our work is indeed probabilistic, and the centers are uniformly located across the sample. We have fabricated many devices to increase the probability of overlapping. For the dipole orientation, we found two preferential directions in the polarization of the single centers. Hence, we have oriented the PhC cavity such that its polarization is aligned with the peak of the distribution of the centers' polarization shown in the supplement. To further scale up the devices, more deterministic positioning will be needed.

Following this comment from the reviewer and to avoid any ambiguity, we replaced "To overcome this challenge in our platform" by "To increase the overlap probability in our platform" in paragraph 2 of page 3.

Referee #1 (Remarks to the Author):

2. In PL measurement, the cavity resonance is observed as a broad resonance at 1272 nm. Is this from some broadband fluorescence from silicon? Please clarify.

Response of the authors to Referee 1

We thank the reviewer for this question. The photoluminescence signal attributed to the resonance of the cavity is due to background photoluminescence. The excitation is done above the gap of silicon, and many defects are pumped that emit a broad spectrum below the gap.

Following this question from the reviewer, we have added a sentence to the manuscript. Please see line 6 on page 4.

The cavity's photoluminescence originates from the broad spectrum of the background centers.

Referee #1 (Remarks to the Author):

3. For the measured ZPL linewidth of 8.3 GHz, is this from the instrument limit or the intrinsic single-center linewidth?

Response of the authors to Referee 1

We thank the reviewer for this question. The measured linewidth is the convolution of the instrument response and the ZPL of the single center. Following the reviewer's input we performed a deconvolution and found that the linewidth is 6.8 GHz. The instrument limit has been added to Figure 3b of the main manuscript (also presented below).

Following this question from the reviewer, we have updated the caption of Figure 3 and added the following update to the main text in page 4, paragraph 2, line 3.

The ZPL is located at 972.43 meV and has a linewidth of 6.8 GHz (obtained after deconvolution with the spectrometer response function).

Fig. 3| **Quantum coherence measurements of the emitter in the cavity.** a, Spectrum of the quantum emitter over a broad range of energy showing the zero-phonon line (ZPL) of the silicon emissive center and its phonon sideband. b, Spectrum of the quantum emitter using a high-resolution grating. The ZPL is located at 972.43 meV and has **an intrinsic linewidth of 6.8 GHz. The dashed line corresponds to the instrument limit.** c, Second-order autocorrelation measurements of the emission from the cavity under continuous excitation. The antibunching at zero delay confirms the successful spatial overlap of a single silicon emissive center with the nanophotonic cavity with an antibunching at zero delay $g^2(0) = 0.30 \pm 0.07$. d, Second-order autocorrelation measurements under pulsed excitation at a repetition rate of 10 MHz. Autocorrelation measurements are performed using a Hanbury-Brown and Twiss interferometer with superconducting nanowire single-photon detectors (see Supplementary Information).

Referee #1 (Remarks to the Author):

4. Following pulsed excitation measurement of G center, the authors claimed on demand single photon generation “Autocorrelation measurements under pulsed excitation at a repetition rate of 10 MHz are presented in Fig. 3d and they demonstrate on-demand single-photon generation from the all-silicon platform”. However, two important criteria for on-demand single photons are

brightness and purity, which the authors did not discuss. Therefore, I think the authors should remove this claim. In addition, throughout the text, the authors did not mention the exact count rate of the Purcell-enhanced G center.

Response of the authors to Referee 1

We agree with the reviewer. Following the reviewer recommendation, we removed the claim related to on-demand single photon generation.

We also added the following sentence to the manuscript and updated the caption of Fig. 3 accordingly. Please see page 5 in the paragraph before the conclusion:

The emission from the ZPL of the center not enhanced is about 700 counts/s. We measured a Purcell enhancement of over 30 with a count rate of 20000 counts/s from the ZPL.

Referee #1 (Remarks to the Author):

5. For g^2 measurements, the baseline at non-zero delay is fitted to exactly 1. How was the g^2 data normalized? I would expect some bunching from the shelving state (3A state shown in Fig 1B). In addition, the timescale in Fig 3c seems very fast compared to the 50 ns lifetime. Was the measurement performed with a very high excitation power? Please clarify.

Response of the authors to Referee 1

The g^2 data was normalized using the following expression:

$$g^2_{norm}(\tau) = g^2_{exp}(\tau) / \Delta \cdot R_1 \cdot R_2 \cdot T,$$

where, Δ is the bin width, R_1 , and R_2 are the count rate of the two detectors, respectively, and T is the total acquisition time.

We excited the center with relatively high power (about $3.3P_{saturation}$) to get the maximum counts. At high power, there is also a repumping mechanism that attenuates the bunching due to the metastable state [Please also see Nature Electronics Redjem *et al.*].

Following this question from the reviewer, we have added the above paragraph to the supplementary file in section 3.1.

Referee #1 (Remarks to the Author):

6. For SIMS measurement (Fig. S7), there are two maximum of carbon concentration at around 0 nm and around 210 nm. Does the author know the origin of these local maximum? Could these be from carbon diffusion during annealing?

Response of the authors to Referee 1

We thank the reviewer for this question. The spikes in ^{13}C concentration plotted at the interface Si/SiO₂ (210nm) is an artefact due to ion yield change caused by materials composition change, which is common in SIMS measurements. At the interface Air/Si we can see a larger concentration of Carbon within a depth of 20nm. The increased concentration close to the surface is due to diffusion of species during annealing at room temperature. The annealing at 1000C that we perform does not seem to affect the carbon profile concentration because the concentration follows very well the SRIMS calculations red line that was done without considering the annealing process.

Following this question from the reviewer, we have added the following to the caption of figure S7 in the supplement section 2.1.

^{13}C concentration spikes at 210 nm is an artefact at the interface Si/SiO₂ due to ion yield change caused by the composition change in materials. The increased concentration close to the surface is due to diffusion of species during annealing at room temperature. The annealing at 1000C that we perform does not affect the carbon profile concentration after implantation at 36keV, because the concentration follows very well the SRIMS calculations (red line) that was done without considering the annealing process.

We would like to thank the reviewer for the quality of his/her questions that have significantly improved the quality of our manuscript. We are happy to implement any additional changes the reviewer thinks are necessary.

Referee #2 (Remarks to the Author):

The authors integrate G-center in silicon photonic crystal cavities and obtain 30-fold Purcell enhancement and 8-fold lifetime reduction of the ZPL single-photon emission at 1275 nm. The results are exciting because of the scalability of the silicon on insulator approach and the authors calibrated the annealing process to optimize single emitter generation and minimize spectral broadening. It is an important work, however, I do not find that it has innovative enough aspects for publication in Nature Communications, but would rather expect to read these findings in Nano Letters or ACS Photonics. Earlier this year Purcell enhancement of G-center ZPL in microrings has been reported in Applied Physics Letters, for example. There is also exciting work on indistinguishability of photons from T-centers and spin control of defects in silicon.

A few comments on the manuscript:

- the temperature of the experiment is missing
- 'quantum photon' is an unusual term since photon is already a quantum of light, perhaps use 'single-photon' or 'quantum light' instead

Response of the authors to Referee 2

We thank the referee for his/her review and for finding that our manuscript is an “important work.” We also want to thank the reviewer for saying that the “results are exciting.”

The reviewer is referring to a paper published earlier this year (already cited in our manuscript as refences Ref 28) that shows the Purcell enhancement of G-centers ensemble using microrings. The claim of our work and the challenge for the community has been to demonstrate the coupling of a single center to a cavity. Our work is the first such demonstration.

Following the reviewer's comments, we have added the temperature in page 4, first paragraph (see highlighted text) and also in the caption of Figure 2(a) of the main the manuscript.

We would like to thank the reviewer for the quality of his/her questions that have significantly improved the quality of our manuscript. We are happy to implement any additional changes the reviewer thinks are necessary.

Referee #3 (Remarks to the Author):

The paper of Redjem et al. reports integration of single G-centers with a 2D photonic crystal cavity to utilize cavity-enhanced light-matter interactions, achieving 30-fold luminescence enhancement and 8-fold emission acceleration. Although using nanophotonic cavity to Purcell-enhance the fluorescence had been demonstrated in multiple solid-state atomic systems and single G-centers in waveguide had been observed before, there is still incremental novelty to analyze single G-centers coupled with the cavity. In large parts, the paper is well-written with well-presented data and convincing conclusions. Thus, I can recommend publication in Nature Communications if the authors can address following concerns and questions:

Response of the authors to Referee 3

We would like to thank the reviewer for finding that “the paper is well-written with well-presented data” and for saying that “I can recommend publication in Nature Communications...”. We also thank the reviewer for the quality of his/her comments towards improving the manuscript. In what follows, we address the comments of the reviewer.

Referee #3 (Remarks to the Author):

1. The ZPL of the single G-center seems to have a rather wide distribution (e.g., Fig. S4, s10) after the generation process including the ion implantation and annealing. For developing quantum sources, e.g., indistinguishable photons from two G-centers, the wide inhomogeneous distribution can be problematic. Can authors comment on the origin of this wide ZPL distribution, and potential ways to mitigate/decrease it?

Response of the authors to Referee 3

We thank the reviewer for this question. After implantation and prior to annealing, the sample shows a high fluence of luminescence which we refer to as “ensemble.”. The density of Carbon-Carbon (CC) pairs is high enough that we always observe many CC’s within the field of view. As the sample is annealed, many color centers “dissolve,” and the density decreases to the point where individual centers can be isolated. However, we would like to emphasize that in our study, the ensemble of G-centers is not just a collection of single G-centers. First, the single center's distribution peak is blue-shifted from the G-line of the

ensemble. When varying the annealing time, we didn't observe a continuous shift of the ZPL peak distribution. Next, the inhomogeneous broadening of the single centers is much larger than the one for the ensemble.

To understand the origin of the broadening and shifting of the ZPL statistics, we performed ab initio calculations and computed the photoluminescence spectrum of the G-centers under strain [Please see Opt. Express 31, 8352-8362 (2023)]. We found out that the volume expands by 1.22% for the isolated G-center as compared to the volume in the presence of a nearby G-center. According to our calculations, the expansion of the lattice corresponding to tensile strain would lead to a blue shift of the ZPL. Thus, the lattice is under high compression after implantation, and annealing releases some of the strain. However, since we used a very short annealing time, the lattice is not fully healed, and the annealing would create more local disorder. This is because annealing silicon results in several microscopic processes that can affect the defect emission locally. For instance, we can have the release of Carbon interstitials from a G-center, the release of silicon interstitials and the formation of vacancy near a G-center, and/or incorporation of carbon interstitials into substitutional lattice positions near a G-centers.

In the three processes discussed above, the silicon lattice locally contracts by a percent or less, causing and leading to an overall "blue-shift" of the emission. Moreover, the magnitude of the induced local strain is highly dependent on the distance between the G-center and the location of the microscopic process causing the strain (see Supplement 1). We notice that the strain can vary significantly depending on the location of the vacancy and may cause the observed large inhomogeneous broadening in the emission of single centers [Opt. Express 31, 8352-8362 (2023)].

Increasing the temperature of the sample necessarily leads to more broadening. Annealing is currently the main step limiting the scalable manufacturing of color centers in silicon. It is possible to find annealing conditions that would prevent the occurrence of such large broadening, such as by using low temperatures and longer annealing time. But it seems that the most promising way is not to use annealing at all and use low fluence implantation instead. However, when using low fluence implantation and no annealing, other challenges arise. Low implantation cannot prevent the formation of an ensemble of centers during the dry etching process. In our work, we use annealing to control the density of single centers and to thermally destroy the centers formed during the dry etching.

Following this question from the reviewer, we have added a sentence to the manuscript. Please see paragraph 1 in page 3 (highlighted section):

Rapid thermal annealing is an important step to thermally cure the broad luminescence from W-centers and G-centers induced by the dry etching process (see Supplementary Information).

Referee #3 (Remarks to the Author):

2. Another important measure of the quantum light source is the photon extraction efficiency, which is a common issue for many solid-state emitters. With the coupling regime authors used, what is the photon extraction efficiency?

Response of the authors to Referee 3

We thank the reviewer for this question. We estimated the total photon collection efficiency ($\eta_{\text{extraction}} \times \eta_{\text{objective}}$) by performing finite-difference time-domain simulations. The extraction efficiency (the first term) was obtained by placing a single dipole within the cavity and calculating the percentage of photons emitted into the far field. The objective-related efficiency (the second term) was determined by the numerical aperture of the objective lens. The resulting total collection efficiency as a function of the numerical aperture is shown in the figure below. In our experiment, with NA=0.85, the total extraction efficiency was approximated as 30%, meaning that 30% of the photons emitted by the coupled emitter (with a near-unity coupling efficiency) were collected by the experimental setup.

Following the question from the reviewer, we have added the Fig. R4 (S15) and the discussion above to the supplementary information file section 3.2 (see highlighted sections in supplement).

Figure R4: Collection efficiency as a function of the numerical aperture of the objective lens.

Referee #3 (Remarks to the Author):

3. It's quite misleading when authors mentioned (in line 56, 57) they can manufacture G-centers with controlled dipole orientations. From Figure S6, G-center dipole orientation after fabrication is random, but with preferences, which enables cavity polarization to match with the G-center dipole. Authors may want to rephrase those sentences to clear this confusion in the main text.

Response of the authors to Referee 3

We thank the reviewer for this comment, and we agree. The experiments revealed that the single center has a preferential direction. We have oriented the PhC cavity such that its polarization aligned with the peak of the distribution of the centers' polarization shown in the supplement. Following the comment from the reviewer, we modified the sentence in page 2, paragraph 2, line 10 as follows.

“The manufacturing of the centers in silicon-on insulator substrates, with controlled densities and preferential dipole orientations, increases their overlap probability with designed nanophotonic cavities.”

Referee #3 (Remarks to the Author):

4. What is the average distance between G-centers in the SOI sample? How many G-centers will locate in the mode volume of the L3 cavity? Does the large $g^2(0)$ being limited by the detection noise (e.g., dark counts) or by background G-center emissions?

Response of the authors to Referee 3

We thank the reviewer for these questions. The density of single quantum emitters, estimated under the annealing condition of 5 s at 1000°C, is $\rho_{QE} = 0.06 \mu\text{m}^{-2}$ corresponding to a single-center effective area of $1/\rho_{QE} = A_{QE} = 16.6 \mu\text{m}^2$. The surface area of the cavity is $A_{cav} = 0.4 \mu\text{m}^2$, which gives a probability of finding a single quantum emitter within the cavity region of $A_{cav}/A_{QE} = 2.4\%$. The probability of having two emitters coupled to the same resonator is negligible.

$g^2(0)$ is currently limited by the PL background. The background may originate from reminiscent defects, or “nonradiative” defects such as the A-form of the G-center. We think that the above gap excitation also contributes to the large background signal because we are exciting all the defects below the gap of the silicon.

In future experiments, one could consider quasi-resonant excitation and better spectral filtering to excite the single emitters and get a better $g^2(0)$, as it is currently done for quantum dots.

Following these questions from the reviewer, we have added a sentence to the manuscript in page 4, end of the second paragraph:

The value at zero delay is mainly limited by the emission from background centers.

Referee #3 (Remarks to the Author):

5. In the G-center decay measurements, what is the instrumentation response function? Authors may just want to add that in Fig. 4c.

Response of the authors to Referee 3

We thank the reviewer for this question. Following the reviewer's comment, the response function of the instrument has been added in the supplementary file section 3.1 and a sentence has been added to the caption of the Fig. 4(c) in the manuscript.

Figure R5: The response function from the time correlated instruments containing the time controller and the Superconducting nanowire single photon detectors. The time response is about 500ps. The function was measured by sending a ultrafast laser pulse (100fs at 1300nm) to the detectors.

Referee #3 (Remarks to the Author):

6. To verify the fluorescence decay enhancement is indeed coming from the emitter-cavity interactions, authors may want to present detuning dependent Purcell factor data and fit it with the Lorentzian (Equation 2 in SI), and to check whether the fitted linewidth matches with the cavity.

Response of the authors to Referee 3

Following the reviewer's comment, the experimental Purcell factor (circle) calculated from the expression is presented in the figure below:

$$F_p^{exp} = (\tau_{bulk}/\tau(\delta) - \tau_{bulk}/\tau_{OFF})/\eta,$$

Where $\tau(\delta)$ is the life as a function of the detuning, and the other terms are explained in the text. The black solid line corresponds to a Lorentzian of the form,

$$F_p^{theory} = 3Q/(0.66 \times 4\pi^2) \cdot \xi^2 \cdot \frac{\Delta\omega_{cav}^2}{4\delta^2 + \Delta\omega_{cav}^2}$$

where Q is the measured quality factor of the cavity, δ is the detuning of the cavity resonance, and $\Delta\omega_{cav}$, is the linewidth of the resonance of the resonator. ξ is the normalized dipole orientation factor (equation S6). In the figure below, the corresponding black line was plotted with a value of $\xi = 0.28$. The center is not positioned at the maximum of the optical field.

We clearly see that equation S3 (black line) matches well with the experiments and proves that the lifetime reduction is due to Purcell enhancement.

Figure R5: Purcell factor as a function of detuning. The black line is the theory (equation S3) and the orange sphere is the experimental data extracted from the equation in the main text.

Following this question from the reviewer, we have added section 3.3 to the supplementary information. We have also added the following to the manuscript, please see page 5, paragraph 1, line 10:

We further confirmed that the Purcell enhancement is due to the cavity-emitter interaction and present in the supplementary detuning-dependent Purcell factors in the Supplementary Information.

Referee #3 (Remarks to the Author):

7. The deposition of the Argon ice seems to decrease the cavity Q factor (i.e., increase the cavity loss rate κ) as shown in Fig. S12c. Can authors comments on this, e.g., whether this comes from increased scattering or perturbation of the band structure? This increasing κ may cause complications on the fitting mentioned in the previous question.

Response of the authors to Referee 3

We thank the reviewer for this question. The Q-factor decreases with the thickness of Argon because of the smaller index contrast between the cavity and the environment compared to air. It is a perturbation that is consistent with our numerical simulation of the photonic cavity by varying the thickness of the Argon layer on top of the cavity.

The quality factor of the cavity drops by 10% for the cavity detuning of less than 6nm that we used in our experiment. This small change in Q has not a significant impact on the fitting that was performed in Figure R5.

Following this question from the reviewer, we have added Figure R6 to the supplementary file section 3.4.

Figure R6: Measured and computed evolution of the quality factor of the nanophotonic cavity as the resonance wavelength is tuned.

We would like to thank the reviewer for the quality of his/her questions that have significantly improved the quality of our manuscript. We are happy to implement any additional changes the reviewer thinks are necessary.

REVIEWERS' COMMENTS

Reviewer #1 (Remarks to the Author):

In their revised manuscript, the authors added new measurements and discussions to address the concerns I raised in the first round of review. The clarity of the manuscript is significantly improved, and all issues have been addressed. Therefore, I recommend the manuscript for publication in Nature Communications.

Reviewer #3 (Remarks to the Author):

I am satisfied that the authors have addressed my concerns and have modified the manuscript in accordance with the suggestions from the reviewers. I would be happy to recommend publication of this work in its current form.